# AN INFORMATION THEORY OF COMPUTE-OPTIMAL SIZE SCALING, EMERGENCE, AND PLATEAUS IN LANGUAGE MODELS

## ABSTRACT

Recent empirical studies show three phenomena with increasing size of language models: *compute-optimal size scaling*, *emergent capabilities*, and *performance plateauing*. We present a simple unified mathematical framework to explain all of these language model scaling phenomena, building on recent skill-text bipartite graph frameworks for semantic learning. Modeling the learning of concepts from texts as an iterative process yields an analogy to iterative decoding of low-density parity check (LDPC) codes in information theory. Thence, drawing on finite-size scaling characterizations of LDPC decoding, we derive the compute-optimal size scaling (Chinchilla rule) for language models. Further, using tools from random network theory, we provide a simple explanation for both emergence of complex skills and plateauing of performance as the size of language models scale. We see multiple plateaus.

## 1 INTRODUCTION

To optimally use computational resources when training language models, several recent studies have empirically investigated how model size and dataset size should scale with compute budget (Kaplan et al., 2020; Hoffmann et al., 2022), finding a certain *allometric rule* much like in mathematical biology (Thompson, 1917; Haldane, 1926). As the sizes of language models continue to increase, large improvements in performance have been observed in certain complex tasks with only a small improvement in the model's loss (Wei et al., 2022) (but see Schaeffer et al. (2023)). The larger language models are therefore said to exhibit *emergent capabilities*, a term drawn from statistical mechanics, where small changes in a macroscopic variable of the system (such as temperature) around a critical value cause an abrupt change—a phase transition or emergent behavior (Baxter, 1982)—in its properties. More recently, there has been prevalent discourse in the AI community that further increases in language model size lead to *plateauing* of performance (Byrnes, 2023; Ritter & Lu, 2024). Although, there have been attempts to explain one or two of these empirical phenomena, a unified mathematical framework that explains all three of these empirically observed phenomena is lacking.

Here we take an approach that builds on information and coding theory (McEliece, 2002) that does so, and also predicts multiple plateaus. In particular, we draw on mathematical ideas around low-density parity check (LDPC) codes (which achieve Shannon optimality) (Sourlas, 1989; Richardson & Urbanke, 2008) and random graph theory (Barabási, 2016). Though statistical language modeling and information theory were introduced in the same paper (Shannon, 1948), modern connections between the two are still fairly limited, cf. Basu et al. (2023).

To provide simple and insightful explanations of empirical phenomena, several abstract frameworks have been proposed (Arora & Goyal, 2023; Liao et al., 2024; Michaud et al., 2023), all based on a skill-text bipartite graph that operates at a semantic level and captures key real-world properties (Yu et al., 2023a). Arora & Goyal (2023) explain emergent phenomena by assuming a compute-optimal size scaling rule (Chinchilla allometry rule) (Hoffmann et al., 2022). Liao et al. (2024) also assume compute-optimal (Chinchilla) size scaling to explain emergence. Michaud et al. (2023) assume power-law scaling and that each text piece contains only one skill, which may be very different than real-world scenarios. Moreover, inverse polynomial loss scaling is interpreted as the average

behavior of emergence at different scales. These existing frameworks explain neither the Chinchilla rule nor the plateau phenomenon. These three frameworks abstract the gradient dynamics of language model training (Arora & Goyal, 2023); an alternate mathematical framework considers dynamics to explain the Chinchilla rule and loss function plateaus but does not consider emergence (Bordelon et al., 2024).

Our information-theoretic approach is inspired by skill-text bipartite graph frameworks of Arora & Goyal (2023); Liao et al. (2024); Michaud et al. (2023) and is closest to Liao et al. (2024). We make a small modification by separating notions of concepts and skills, as in well-established human cognitive architectures (Newell, 1990) that have simple hierarchies (Laird et al., 1987; Anderson, 1993; Kieras & Meyer, 1997). In our framework, skills are not directly learned from text; rather, concepts are learned from texts and skills at different levels are learned from concepts (see Section 2 for a detailed description). That is, our framework takes on the notion of skill-quanta from Michaud et al. (2023), and so the number of concepts a language model can learn is proportional to the model size (number of model parameters).

The key difference in our work is to have much more detailed and expressive analysis using non-asymptotic techniques rather than asymptotic ones (Di et al., 2002). Indeed, such finitary analysis is necessary to even consider size scaling. Recall that Arora & Goyal (2023); Liao et al. (2024) assume Chinchilla scaling, whereas we derive it without it being built into our framework.

The main contributions of this paper are as follows.

1. We propose a simple unified mathematical framework that considers a language model's learning of concepts from texts and composition of skills from concepts.
2. Using this framework and tools from non-asymptotic information theory, we deduce compute-optimal scaling in language models.
3. With the help of random network theory, we provide a simple explanation for emergent abilities of language models in complex tasks when their sizes exceed a certain threshold.
4. We show that plateauing of performance with size-scaling is just a consequence of diversity of skills required for a task. Moreover, plateauing indicates the possibility of multiple emergences as language models continue to scale further.

Our work takes a step in grounding empirical phenomena observed in size scaling of language models on a rigorous mathematical footing. Understanding the origin of these phenomena may yield insights into better architectures, better datasets, and the limitations of large-scale learning systems. Provable optimality of the Chinchilla rule (as in Proposition 1), however, may indicate that there are no gains from better scaling of data and compute remaining, cf. Ho et al. (2024, Appendix B). Separately, our results may help policymakers develop regulatory policy by providing insight into the relationship between capabilities of concern and controllable resources such as data and compute (Hooker, 2024).

## 2 GRAPH-BASED FRAMEWORK

Our framework is based on the notion of learning as two levels. First, a set of concepts are learnt from a set of texts with each text involving one or more concepts. Second, learning concepts enables the language model to acquire skills, and after encountering a sufficient number of texts with co-occurring pairs of skills, the model eventually acquires compositional abilities resulting in emergent phenomena in various complex tasks. The framework naturally leads to information-theoretic analysis in Section 3.

### 2.1 TEXTS, CONCEPTS, AND SKILLS

A set of tokens constitute a text (similar to a text piece defined in Arora & Goyal (2023)) from which a language model can learn a set of concepts. This is modeled as a concept-text bipartite graph similar to the skill-text bipartite graph in Liao et al. (2024). Note that we consider single epoch training as described in Hoffmann et al. (2022). The total number of concepts a model can learn depends on its size (number of model parameters). Here we consider a hierarchy of skills: basic skills in the first layer and multiple layers of advanced skills. Basic skills are easily acquired from concepts, whereas acquiring advanced skills additionally requires certain prerequisite (less advanced) skills. We formalize these semantic learning notions in the sequel. Note that this bipartite graph formulation

of learning is intimately related to graph-based approaches to data compression (Martinian & Yedidia, 2003) and associative memory (Karbasi et al., 2013). Moreover, although this approach to abstract modeling has been tied to Transformer-based language modeling architectures (Yu et al., 2023a), it can describe a variety of quite different learning paradigms (Yu et al., 2023b).

## 2.2 NOTATION

Let $\mathcal{T}$ be a subset of text pieces from a set $\mathfrak{T}$, and let $\mathcal{R}$ be a subset of concepts from a set $\mathfrak{R}$. Let the model size $N$ (number of parameters) be proportional to the number of concepts $R = |\mathcal{R}|$, i.e., $N = \varsigma R$, for some $\varsigma > 0$.[1] Similarly, let $\tau$ be the number of tokens in a text piece $t \in \mathcal{T}$ with $T = |\mathcal{T}|$, implying that the dataset size $D = \tau T$. For a given compute budget $C$,[2] a language model of size $N$ can be trained using a dataset of size $D$ so the constraint $6ND \leq C$ is satisfied (see Hoffmann et al. (2022)).

Correspondingly, for a given compute budget, $G_1^{(C)} = (\mathcal{T} \cup \mathcal{R}, E_{\mathcal{T}\mathcal{R}})$ denotes a concept-text bipartite graph, where an edge $e_{tr} \in E_{\mathcal{T}\mathcal{R}}$ indicates that the language model can learn concept $r$ from text $t$. Let the degrees of text pieces (number of skills required to understand a text) be binomially distributed with a fixed mean degree $d_t$, i.e., $P_R = \text{Binomial}(n, p) = \text{Binomial}(R, d_t/R)$. The corresponding generating function is $P_R(x) = \sum_i P_i x^i$, where $P_i = \Pr(Y = i)$ and $Y \sim \text{Binomial}(R, d_t/R)$. Let the degree distribution of concepts be $L_T = \text{Binomial}(T, d_r/T)$, where $d_r = d_t T/R$. Note that $d_t/R = d_r/T =: p$. There is an alternate point of view: If we assume that there exists an edge between a text piece and a concept with probability $d_t/T$, then a typical graph will have text and concept degree distributions close to $P_R$ and $L_T$, respectively. It is generally useful to view degree distribution from an edge-perspective, which is $\lambda_T(x) = L_T'(x)/L_T'(1)$ and $\rho_R(x) = P_R'(x)/P_R'(1)$ (Richardson & Urbanke, 2008).

Let $G_2 = (\mathfrak{R} \cup \mathcal{S}, E_{\mathfrak{R}\mathcal{S}})$ be a skill-concept graph, where $\mathcal{S} = \cup_l \mathcal{S}^{(l)}$ denotes a set of hierarchical skills, with finite number $S^{(l)}$ of skills in each level $l$. Each concept is connected to a unique skill at every level $l$, i.e., each concept enables learning of one skill at each level, and each skill $s^{(l)}$ is connected to $\sigma_l$ prerequisite skills at level $l - 1$. Our unified framework is represented by the graph $G^{(C)} = G_1^{(C)} \cup G_2$ as shown in Figure 1.

## 2.3 LEARNING CONCEPTS FROM TEXT PIECES

Following the approach described in Liao et al. (2024), we assume that a language model learns concepts from text pieces as an iterative peeling process. For a self-contained explanation, let us briefly describe the peeling process here. Let $\mathcal{R}_+^{(u)}$ denote the set of concepts learnt, and $\mathcal{R}_-^{(u)}$ denote the set of concepts not learnt in peeling iteration $u$. Initially, all the concepts are unlearned, i.e., $\mathcal{R}_-^{(0)} = \mathcal{R}$ and $\mathcal{R}_-^{(0)} = \emptyset$. Next, a language model learns a concept $r \in \mathcal{R}_-^{(0)}$ if a text piece $t \in \mathcal{T}$ is uniquely connected to $r$ yielding $\mathcal{R}_+^{(1)} = \{r\}$ and $\mathcal{R}_-^{(1)} = \mathcal{R}_-^{(0)} \setminus \{r\}$. Before the next iteration, the edge $e_{tr}$ and concept node $r$ from the graph are removed. The next iteration starts by finding another text piece uniquely connected to a concept in $\mathcal{R}_-^{(1)}$, and this peeling process continues until there is either no more text piece/s connected to a unique concept in $\mathcal{R}_-$ or all the concepts are learnt, i.e., $\mathcal{R}_+ = \mathcal{R}$.

## 2.4 ACQUISITION OF SKILLS AND COMPOSITION OF SKILLS

A skill $s^{(l+1)}$ at level $l+1$ is considered acquired when two conditions hold: 1) all the $\sigma_{l+1}$ prerequisite skills at the lower level $l$ are learnt, and 2) at least one concept associated with $s^{(l+1)}$ is learnt. A pair of concepts $(r_1, r_2)$ is considered connected (denoted by $r_1 - r_2$) if there is a path $r_1 - t - r_2$ through at least one text $t \in \mathcal{T}$. Then, for a fixed level $l$, a skill-graph $G_2^{(l)} = (\mathcal{S}^{(l)}, E_{\mathcal{S}^{(l)} \times \mathcal{S}^{(l)}})$ is constructed as follows: A pair of skills $s_1$ and $s_2$ in $\mathcal{S}^{(l)}$ has a direct link (i.e., $e_{s_1 s_2} \in E_{\mathcal{S}^{(l)} \times \mathcal{S}^{(l)}}$) if there are at least $\eta_l$ distinct paths $s_1^{(l)} - r_1 - r_2 - s_2^{(l)}$ (with at least $\eta_l$ distinct pairs of concepts $(r_1, r_2)$), and all the $2\sigma_l$ prerequisite skills required for both skills are acquired. The intuition behind

---

[1]Here, a *concept* is similar to a *skill quantum* in Michaud et al. (2023).

[2]Compute budget is measured in number of floating point operations or FLOPs (Hoffmann et al., 2022).

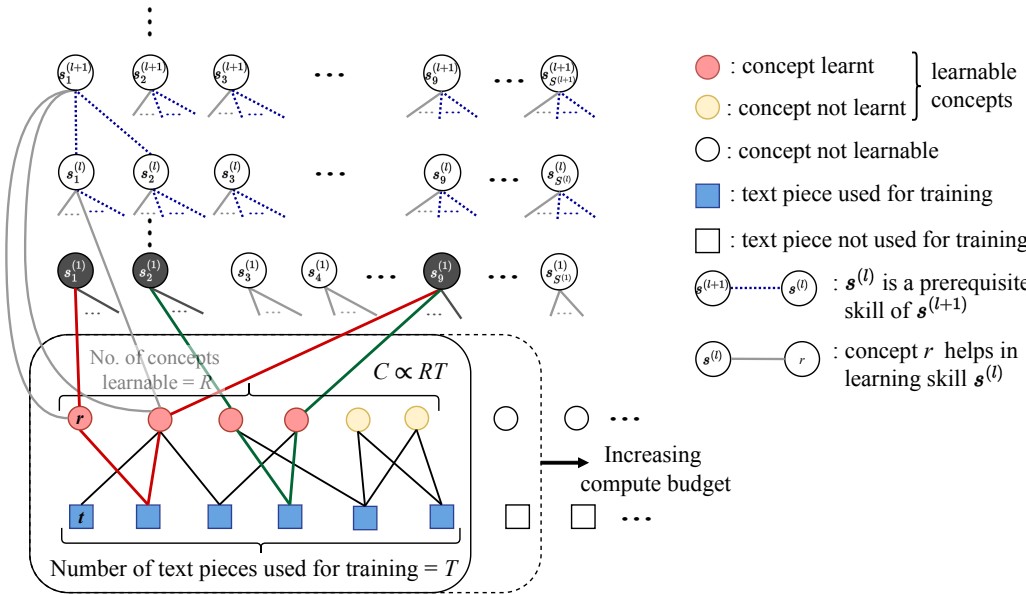

Figure 1: A unified framework of learning concepts and skills by language models. The lower subgraph $G_1^{(C)}$ is a concept-text bipartite graph akin to a Tanner graph representation of an LDPC code. The upper subgraph $G_2$ shows concept-skill and skill-to-skill relationships, with multiple levels of skills denoted by $l$. Higher $l$ indicates more advanced skills.

this construction is that a pair of skills is connected (and therefore can be composed) if they co-occur sufficiently many times through distinct pairs of concepts in the training data, and all prerequisite skills of both skills are already acquired. Further, since more advanced skills are generally hard to learn, skills at higher levels (larger values of $l$) need larger values of $\eta_l$.

## 2.5 DEFINING EMERGENCE

In the context of neural language models, there are several definitions of skill emergence in the literature. In most existing frameworks, skills are directly associated with texts. In the skill-text bipartite graph framework of Arora & Goyal (2023), the fraction of text pieces in error (incorrect answers to cloze questions) is obtained from the Chinchilla rule, where the error fraction is smaller for larger model sizes. Emergence is defined in terms of the error rate of the skill-tuple, i.e., the fraction of edges to error-marked text pieces from $k$-tuple of skills, as follows: For a fixed target error threshold, with an increase in model size, emergence is defined as increase in the largest size of the skill tuple $k$ whose error rate is below the threshold. According to this definition of emergence, there is no phase transition and conforms to the notion that emergence is slow.

In Liao et al. (2024), emergence is defined as a function of the ratio of number of text pieces to skills: with increase in the ratio of number of texts to skills, emergence is defined as the increase in the size (normalized) of the largest connected component corresponding to the learnt skills. This definition of emergence exhibits a phase transition around a specific value of text-to-skill ratio. However, this definition of emergence as a function of text-to-skill ratio (not of model size) does not follow the definition of emergence, for example in Wei et al. (2022): "An ability is emergent if it is not present in smaller models but is present in larger models."

A critical view of emergent abilities is given in Schaeffer et al. (2023), arguing that emergence in performance (e.g. in terms of accuracy) is only a consequence of quantization of another metric (e.g. token edit distance) which shows gradual improvement with size scaling, and hence is only a mirage. Although this argument holds, we maintain a more optimistic view. How the performance of the model is measured is important, but corresponding to a performance metric, there is an abstract quantity such as the ability to compose multiple skills, which a language model gains when the model size exceeds a certain threshold to exhibit a true phase transition.

In our framework, advanced skills (larger $l$) are acquired from concepts and more basic skills, rather than directly from text pieces. To describe the composition of skills not seen in training, we begin by asserting transitivity of skill composition for a fixed skill level $l$: if the training data contains enough text pieces with composition of both pairs $(s_1^{(l)}, s_2^{(l)})$ and $(s_2^{(l)}, s_3^{(l)})$, then a language model is capable of composing skill $s_1^{(l)}$ and $s_3^{(l)}$. Consequently, a language model successfully performs a sub-task requiring a composition of a set of skills $\mathcal{S}_\theta^{(l)} \subseteq \mathcal{S}^{(l)}$ if there is a path between every pair of skills belonging to $\mathcal{S}_\theta^{(l)}$ in graph $G_2^{(l)}$. As we will see in Section 3.3, the skill graph with nodes $\mathcal{S}^{(l)}$ is an Erdös-Rényi (ER) random graph with edges indicating pairwise composition of skills seen in training, and the relationship between composition of a set of skills and the presence of giant connected component.

For small compute budgets, dataset size corresponding to compute-optimal performance is small, in which case the training data contains composition of only a small number of skill pairs. As compute budget increases, the size of the training data increases, and therefore the number of composed skill pairs seen by the language model during training increases. Beyond a certain compute-budget threshold and due to skill composition transitivity, the ability of the language model to compose most skill pairs emerges, appearing as a phase transition around this compute-budget threshold. As we will see in Section 3.3, this phase transition is related to the appearance of a giant connected component (GCC) in random graphs with increasing edge probability. Our definition of emergence exhibits phase transition as empirically observed in language models, and our finitary analysis helps in conforming to the definition of emergence in Wei et al. (2022).

## 3 Explaining all three phenomena

Using the framework in Section 2, we aim to explain the compute-optimal (Chinchilla) scaling rule by applying non-asymptotic information-theoretic tools to the bipartite graph $G_1^{(C)}$ and to explain emergence and plateauing phenomena based on the density of connections in the skill-graphs $\{G_2^{(l)}\}_l$.

### 3.1 Compute-optimal scaling rule

Let $\mathcal{R}_+ \subseteq \mathcal{R}$ denote the set of concepts learnt after the peeling process terminates. Note that the corresponding number of concepts $R_+ = |\mathcal{R}_+|$ is a random variable. The goal of the language model is to minimize training loss. We assume that the language model inherently attempts to maximize the number of concepts learnt from the text pieces under the compute budget constraint $C$, which yields the following constrained optimization problem.

$$\underset{R,T}{\text{maximize}} \ \mathbb{E}_{G_1^{(C)} \sim (\lambda_T, \rho_R)}[R_+] \tag{1}$$

$$\text{s.t. } RT \leq C',$$

where the number of model parameters $N = \varsigma R$, number of tokens in a text piece is $\tau$, $C' = \frac{C}{6 \varsigma \tau}$, and $(R^*, T^*)$ is the maximizer of the objective function in equation 1. It follows directly that the objective function in equation 1 can be rewritten as:

$$\mathbb{E}_{G_1^{(C)} \sim (\lambda_T, \rho_R)}[R_+] = R(1 - \Pr\{r \notin \mathcal{R}_+ | R, T\}). \tag{2}$$

For a bipartite graph sampled from a degree distribution pair $(\lambda_T, \rho_R)$, one may exactly compute the number of learned concepts using combinatorial arguments (Di et al., 2002). However, the exact analysis becomes computationally expensive very quickly with increasing compute budget $C$ (equivalently $R$ and $T$). Moreover, since we are mainly interested in scaling behavior, the exact analysis may not be very insightful. Fortunately, observing that the peeling process is equivalent to iterative decoding of LDPC codes when the codeword symbols are corrupted by erasure, allows us to sidestep this difficulty. The trick is to construct a parent bipartite graph $\widetilde{G}_1^{(C)}$ with $(1 - \epsilon)R/\epsilon$ additional concept nodes[3] and degree distribution pair $\lambda_T$ and $\widetilde{\rho}_R$, such that the peeling process in

---

[3] $\epsilon \in (0, 1)$ can be chosen arbitrarily, and $\Pr\{r \notin \mathcal{R}_+ | R, T\}$ is invariant to $\epsilon$. See Appendix A.2 for details.

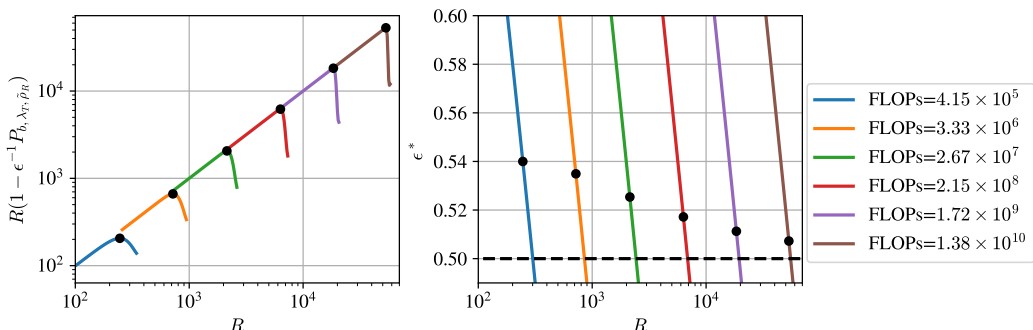

Figure 2: **IsoFLOP curves**: (**left**) Number of concepts learnt as a function of $R$ for different compute budgets (FLOPs); (**right**) Block erasure threshold as a function of the number of concepts $R$ for different compute budget. In both subfigures, solid black markers indicate the points corresponding to $R^*$.

this graph appears as belief propagation decoding of the $\epsilon$ fraction of erased codeword symbols (see Appendix A.2 for details), which yields

$$\Pr\{r \notin \mathcal{R}_+ | R, T\} = \frac{P_{b,\lambda_T,\widetilde{\rho}_R}}{\epsilon}, \tag{3}$$

where $P_{b,\lambda_T,\widetilde{\rho}_R}$ is the post-decoding bit erasure rate corresponding to $\widetilde{G}_1^{(C)}$ (see Appendix A.2 for details).

Before providing an expression for $P_{b,\lambda_T,\widetilde{\rho}_R}$ some notations are as follows: let $f(x,\epsilon) = \epsilon \lambda_T(1 - \widetilde{\rho}_R(1-x))$, then the decoding threshold $\epsilon^* = \inf\{\epsilon \in [0,1] : x = f(x,\epsilon) \text{ has a solution in } x \in (0,1]\}$, $x^*$ be a critical point satisfying $x^* = f(x^*,\epsilon^*)$, $\nu^* = \epsilon^* L_T(1 - \widetilde{\rho}_R(1-x^*))$. Substituting for the post-decoding bit erasure rate $P_{b,\lambda_T,\widetilde{\rho}_R}$, the objective function in equation 1 is given by (see Appendix A.2 for more details):

$$\mathbb{E}_{G_1^{(C)} \sim (\lambda_T, \rho_R)}[R_+] \approx R\left(1 - \frac{\nu^*}{\epsilon} Q\left(\sqrt{\frac{R}{\epsilon}} \frac{(\epsilon^* - \epsilon)}{\alpha}\right)\right), \tag{4}$$

where $\alpha$ depends on the degree distribution pair $(\lambda_T, \widetilde{\rho}_R)$ (see Appendix A.2 for the closed-form expression), and $Q(\cdot)$ is the complementary Gaussian cumulative distribution function. In Figure 2, the objective function in equation 1 is plotted against the number of concepts $R$ for multiple compute budgets. In the left subfigure, each curve corresponds to a fixed compute budget. Note that smaller values of $R$ correspond to smaller language model sizes, in which case the dataset size (number of texts $T$) is more than necessary for the model to learn all the skills. Contrarily, for large model sizes, the smaller dataset size is insufficient to learn the concepts well. There is an optimum model size and dataset size pair (equivalently $R$ and $T$) such that the number of concepts learnt is maximized, as indicated by a solid black marker for each compute budget $C$. This figure is analogous to isoFLOP curves in (Hoffmann et al., 2022, Figure 2), where training loss is plotted against model size for different compute budgets.

Compute-optimal size scaling of model size and dataset size with increasing compute budget obtained by numerically solving equation 1 is shown Figure 3(a). The markers in the figure correspond to the empirically predicted model size and dataset size for compute-optimal performance of the Chinchilla model reported by Hoffmann et al. (2022) when the compute budget is $5.76 \times 10^{23}$. In the following proposition, we prove that the Chinchilla rule is optimal.

*Proposition* 1. **Compute-optimal size scaling**: Suppose the number of model parameters and the dataset size scale with compute budget $C$ as $N \propto C^\alpha$ and $D \propto C^\beta$ for some $\alpha > 0$ and $\beta > 0$. For compute-optimal performance of a language model, the dataset size $(D)$ and model size $(N)$ must scale equally with the increasing compute budget $C$ (or FLOPs), i.e., $\alpha = \beta = \frac{1}{2}$.

*Proof.* The approach is to prove that neither $T/R = o(1)$ nor $R/T = o(1)$ maximizes the objective function in equation 1. This implies that $R/T$ must be a constant, i.e., $R$ and $T$ must scale equally with compute budget $C$.

Denote $\epsilon^*$ be the decoding threshold corresponding to the degree distribution pair $(\lambda_T, \widetilde{\rho}_R)$. The upper bound on the decoding threshold is given by (see (Richardson & Urbanke, 2008, Section 3.14.4))

$$\epsilon^* \leq \frac{\int \widetilde{\rho}_R}{\int \lambda_T} =: \epsilon^*_{ub}$$

(a) If $\frac{T}{R} = o(1)$ (i.e., $\frac{T}{R}$ decays as $C \to \infty$), then

$$\epsilon^*_{ub} - \epsilon \leq \epsilon \left( \left( 1 - e^{-d/\epsilon} + \frac{d^2}{\epsilon R} \right) \left( \frac{1}{d} + \frac{T}{R} \right) - 1 \right) \xrightarrow{C \to \infty} \epsilon \left( \frac{(1 - e^{-d/\epsilon})}{d} - 1 \right) < 0,$$

which implies that $P_{b,\lambda_T,\widetilde{\rho}_R} \to 1$. Therefore, number of skills learnt vanishes for large $C$.

(b) Consider $\frac{R}{T} = o(1)$. From the fixed point characterization of decoding threshold of LDPC codes, we have

$$f(x, \epsilon^*) = \epsilon^* \lambda_T (1 - \widetilde{\rho}_R(1 - x)),$$
$$= \epsilon^* (1 - (1 - xp)^{\frac{R}{\epsilon} - 1} p)^{T-1}, \tag{5}$$

where $p = d_t / R$. Since $R/T = o(1)$, the number of text pieces $T$ grows strictly faster than $R$ with respect to compute budget $C$, implying that the second term in equation 5, i.e., $(1 - (1 - xp)^{\frac{R}{\epsilon} - 1} p)^{T-1} \to 0$ for large $C$. Therefore, for a non-trivial solution, i.e., $x = f(x, \epsilon^*) \in (0, 1]$, the decoding threshold $\epsilon^*$ must be very large. As a result, the post-decoding bit erasure rate $P_{b,\lambda_T,\widetilde{\rho}_R}$ vanishes for large $C$.

Suppose, $(R^*_C, T^*_C)$ such that $R^*_C/T^*_C = o(1)$ minimizes equation 1 (subscript $C$ shows explicitly the dependence on $C$). Now, consider $\hat{R}_C = R^*_C(1 + \delta)$ and $\hat{T}_C = T^*_C/(1 + \delta)$. Note that $\hat{R}_C/\hat{T}_C = (1+\delta)^2 R^*_C/T^*_C = o(1)$. Therefore, for any $\delta' \in (0, \delta)$, there exists $C_0$ such that for all $C \geq C_0$ the bit erasure rate $\epsilon^{-1} P_{b,\lambda_{\hat{T}_C},\widetilde{\rho}_{\hat{R}_C}} \leq \delta'/(1 + \delta')$. Now consider the ratio of number of concepts learnt:

$$\frac{\hat{R}_C(1 - \epsilon^{-1} P_{b,\lambda_{\hat{T}_C},\widetilde{\rho}_{\hat{R}_C}})}{R^*_C(1 - \epsilon^{-1} P_{b,\lambda_{T^*_C},\widetilde{\rho}_{R^*_C}})} \geq \frac{R^*_C(1 + \delta)\left(1 - \frac{\delta'}{1+\delta'}\right)}{R^*_C} = \frac{1 + \delta}{1 - \delta'} > 1, \tag{6}$$

where the first inequality is by substitution and using the fact that $\epsilon^{-1} P_{b,\lambda_{T^*_C},\widetilde{\rho}_{R^*_C}}$ is non-negative, and the second inequality is because $\delta' < \delta$. Therefore, $(R^*_C, T^*_C)$ is not a maximizer, which is a contradiction. Therefore, $R/T$ cannot be $o(1)$.

Since $R/T = c\, C^{\alpha - \beta}$ for some $c > 0$, $\alpha = \beta$ must hold, i.e., $R/T$ must asymptotically be a constant. Noting that $RT \propto C$, we obtain $\alpha = \beta = \frac{1}{2}$. In other words, the model size $N$ and dataset size $D$ must scale equally with compute budget $C$.

$\square$

## 3.2 Scaling of excess entropy

The cross-entropy loss is the sum of two terms, namely, entropy of the ground truth distribution and excess entropy (see Arora & Goyal (2023) for more details). Here, investigate the how a lower bound of excess entropy scales with compute budget according our framework. Under finitary analysis, for every compute budget $C$, there is an associated error rate $P_{b,\lambda_T,\widetilde{\rho}_R}/\epsilon$ which indicates a fraction of concepts are not learnt even after the peeling process is complete. Similar to Arora & Goyal (2023), we assume that cloze questions associated with text pieces connected to unlearnt concepts are incorrectly answered. Therefore, the training error is equivalent to the probability that a check node (text piece) is connected to the stopping set (unlearnt concepts) at least twice. Refer to Richardson & Urbanke (2008) on stopping sets. The training error corresponding to $(N, D)$ given a compute budget $C$ is (see Appendix B for the calculation):

$$P_{e,train} \geq 1 - \left(1 - \frac{d_t P_b}{R}\right)^{R-1} - d_t P_b \left(1 - \frac{d_t P_b}{R}\right)^{R-1} \approx d_t^2 \epsilon^{-2} P_{b,\lambda_T,\widetilde{\rho}_R}^2. \tag{7}$$

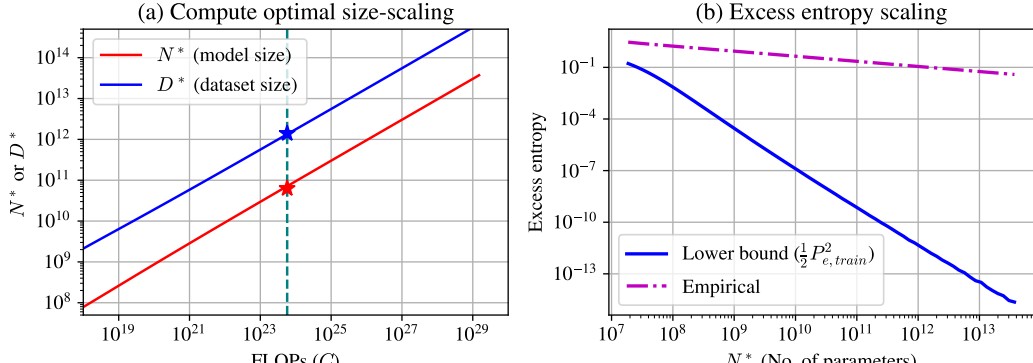

Figure 3: (a) Model and dataset size pair $(N^*, D^*)$ that maximizes equation 1 as a function of compute budget $C$. The curves being parallel in logarithmic scale indicates that model size and dataset size must scale equally with $C$. In this subplot, we set $\varsigma = 2 \times 10^5$, $\tau = 8 \times 10^5$, and $d_t = 6$. The markers indicate $(N, D)$ corresponding to the compute-optimal performance predicted by the Chinchilla rule (Hoffmann et al., 2022) when compute budget is $5.76 \times 10^{23}$ (dashed vertical line); (b) Scaling of the lower bound of excess entropy in equation 8 compared with empirically observed scaling according to (Hoffmann et al., 2022) as a function of the model size $N^*$.

Using Pinsker's inequality that relates Kullback-Leibler divergence to total variational distance as $D_{KL}(P\|Q) \geq \frac{1}{2}\|P - Q\|_1^2$, and the equivalence between total variation distance and error rate on cloze questions (Arora & Goyal, 2023), we obtain the following lower bound on excess entropy:

$$\text{Excess entropy} \geq \frac{1}{2}P_{e,train}^2 \gtrapprox \frac{1}{2}d_t^4\epsilon^{-4}P_{b,\lambda_T,\widetilde{\rho}_R}^4. \tag{8}$$

Empirically observed excess entropy scaling (an upper bound) of transformer-based models and a lower bound according to our framework in equation 8 are depicted in Figure 3(b). The gap between them indicates the scope for either tightening theoretical lower bound or devising architectures that offer better empirical scaling or both.

### 3.3 EMERGENCE

As the model size increases (along with the compute budget $C$) there is a sharp increase in performance (e.g. accuracy) of the language model on certain complex tasks which the model was not trained on. We aim to provide a simple explanation to this empirical phenomenon using random graph theory.

Let $p_l$ denote the probability there is a direct link between any two pairs of skills at level $l$. For a fixed $(R, T)$, $p_l$ evaluates as (see Appendix C for the derivation):

$$p_l \geq \begin{cases} (1 - g(R, p_{rr}, \eta_l)) \gamma_{l-1}^{2\sigma_l} & \text{if } \eta_l \leq \binom{R}{2}p_{rr} \\ \frac{1}{\sqrt{8\eta_l\left(1-\eta_l/\binom{R}{2}\right)}} g(R, p_{rr}, \eta_l)\gamma_{l-1}^{2\sigma_l} & \text{otherwise,} \end{cases} \tag{9}$$

where $g(R, p_{rr}, \eta_l) = \exp\left(-\binom{R}{2}D_{KL}\left(\frac{\eta_l}{\binom{R}{2}}\|p_{rr}\right)\right)$, $p_{rr}$ is the probability that a pair of concepts occur in at least one text piece, and $\gamma_{l-1}$ is the probability that a skill belongs to GCC of $G_2^{(l)}$ (which we show next). Recall the definition of emergence from Section 2.5 as the ability of a language model to compose all pairs of skills within a subset of skills in a given level $l$ required for a specific task. In this regard, note that the skill graph $G_2^{(l)}$ is equivalent to an Erdös-Rényi (ER) random graph with $S^{(l)}$ nodes and edge probability $p_l$. A pair of skills in level $l$ can be composed if there is a path between them in $G_2^{(l)}$, and the probability that there is a path between any pair of skills is bounded below by the probability that both skills are in GCC of $G_2^{(l)}$.

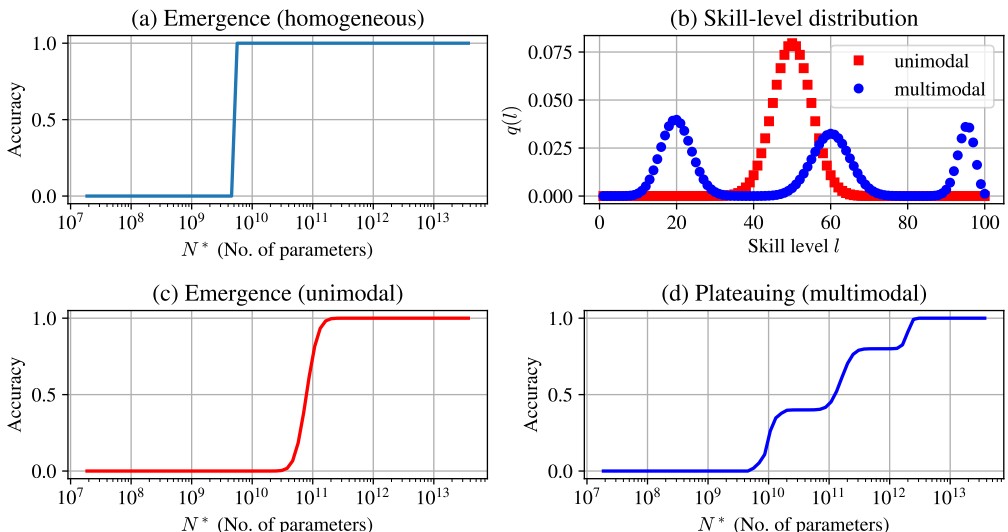

Figure 4: Accuracy of the language model sharply increases after the model size (equivalently $C$) exceeds a threshold, which is a consequence of the emergence of a GCC in a skill graph $G_2^{(l)}$. (a) Step increase in accuracy for a homogeneous task. (b) Skill level distribution $q(l)$ for unimodal and multimodal heterogeneous tasks. (c) Smooth emergence for unimodal heterogeneous task. (d) Plateauing phenomena as a consequence of a task requiring diverse skills according to multimodal distribution. In this subplot, we used the following values for the parameters: number of skill levels $L = 100$, $S^{(l)} = 10^3$, $\eta_l = \exp(7l/L)$, $\sigma_l = \log_2(l)$ for all $l \in \{1, \dots, L\}$, $q(m) = 1/6$ for all $m \in \{2, \dots, 7\}$.

Suppose $\gamma_l$ is ratio of the size of GCC in $G_2$ to the total number of skills (number of nodes in $G_2^{(l)}$) at level $l$, i.e., $\gamma_l = S_{GCC}^{(l)}/S^{(l)}$. Note that $\gamma_l$ is equivalent to the probability that a skill at level $l$ is in GCC. From random graph theory (Barabási, 2016), for an ER graph with edge probability $p_l$, the solution to the following equation yields $\gamma_l$:

$$\gamma_l = 1 - \exp\left(-p_l S^{(l)} \gamma_l\right), \tag{10}$$

where $p_l S^{(l)}$ is the mean degree of the ER skill graph. The solution is

$$\gamma_l = 1 + \frac{1}{p_l S^{(l)}} W_0 \left(-p_l S^{(l)} \exp\left(-p_l S^{(l)}\right)\right), \tag{11}$$

where $W_0(\cdot)$ is the upper branch of the Lambert $W$ function. The ratio $\gamma_l$ has a phase transition at $p_l = 1/S^{(l)}$. To see this, note that $W_0(xe^x) = x$ for $x < -1$. Therefore, whenever $p_l < 1/S^{(l)}$, $\gamma_l$ is identically zero. As $p_l$ increases beyond $1/S^{(l)}$, $|W_0(\cdot)|$ starts decreasing and consequently, $\gamma_l$ increases.

For a particular skill level $l$, $\gamma_l$ and $p_l$ can be computed recursively using equation 11 and equation 9, with the following initial conditions: $\gamma_0 = 1$ and $\sigma_l = 0$ (observe that no prerequisite skill is required to learn basic skills, i.e., skills at $l = 1$). Suppose a task requires $m$ skills at level $l$ (a homogeneous task), the model performs the task successfully only if there is a path between every pair of those skills in $G_2^{(l)}$. Therefore, a sufficient condition is that all skills required for the task are in GCC. The accuracy of the task is:

$$\text{Accuracy} = \Pr\{\text{Composition of } m \text{ skills in } \mathcal{S}^{(l)}\},$$
$$= \Pr\{\text{There exists a path between every pair among } m \text{ skills in } \mathcal{S}^{(l)}\},$$
$$\geq \Pr\{\text{All } m \text{ skills} \in \text{GCC of } G_2^{(l)}\} = \gamma_l^m. \tag{12}$$

The accuracy curve in Figure 4 shows a step phase transition with increasing model size. This is a consequence of the homogeneous task requiring skills at only one level. However, empirically observed accuracy curves exhibit smoother phase transitions (Wei et al., 2022). To demonstrate a smoother phase transition, consider a complex heterogeneous task that requires diverse skills at different levels, in particular consisting of subtasks requiring $m$ skills at level $l$ with probability $q(l, m)$. Task accuracy is:

$$\text{Accuracy} \geq \sum_{l,m} q(l, m) \gamma_l^m. \tag{13}$$

The overall accuracy is therefore a weighted average of the emergence curves. To illustrate this using a numerical example, consider a skill graph $G_2$ with $L = 100$ levels, let $q(m, l) = q(m)q(l)$ with $q(m) = 1/6$ for $m \in \{2, \ldots, 7\}$ and consider a binomial distribution, $\text{Binomial}(L, 1/2)$, over the skill levels, i.e., $q(l) = \binom{L}{l}(\frac{1}{2})^L$ as shown in Figure 4(b). The corresponding accuracy according to equation 13 is shown in Figure 4(c). In general, a smooth single phase transition can be obtained by a unimodal distribution over skill levels with a sufficiently large variance.

## 3.4 Plateauing

According to our framework, plateauing in accuracy after encountering an emergent phenomenon (with scaling) occurs because of the greater diversity of skills (at multiple levels) required by the heterogeneous task under consideration. In particular, we observe plateauing when the skill levels required for a task follows a multimodal distribution. To illustrate this, consider a mixture of binomial distributions over the skill levels, i.e., $q(l) = \sum_i w_i \text{Binomial}(L, \pi_i)$, with $(w_i)_i \in (2/5, 2/5, 1/5)$ and $(\pi_i)_i = (0.2, 0.6, 0.95)$ is shown in Figure 4(b). The corresponding accuracy according to equation 13 is shown in Figure 4(d). In general, a multimodal distribution over skill levels results in emergence at multiple scales and plateaus between them. Our framework yields an interesting trend associated with the plateauing of performance: plateauing indicates the possibility of one (or more) upcoming emergent phenomenon (phenomena), which one would encounter with further scaling.

## 4 Conclusion

We presented a simple unified framework to explain all three empirical phenomena observed with size scaling of language models. Existing frameworks assume a compute-optimal scaling rule and only then explain emergent phenomena. We use non-asymptotic information theory to explain both compute-optimal size scaling and emergent abilities of language models. Moreover, we explain the more recent empirical phenomenon of plateauing of performance using random network theory, and also predict that plateauing implies the possibility of multiple emergent phenomena with further size scaling.

There are some open questions and considerations worth exploring. We do not take training time into account in our framework. Therefore, we do not explain (or attempt to explain) empirical phenomena such as double descent or grokking (Huang et al., 2024). Perhaps future work can either incorporate training epochs in our framework or propose a different novel framework to explain them. Even though the sequential learning of concepts through peeling process gives certain ordering to concepts, there is no inherent ordering of concepts and we do not consider concept hierarchies (Yu et al., 2023b;c). One can explore the advantages of doing so. Evidently, the degree distribution of texts is related to the model's architecture. Therefore, optimizing the degree distribution enables a language model to learn more concepts from text pieces. Further, the quality of the training data is related to text-to-concept edge deletions in sequential concept learning, which can be incorporated into our framework. Such optimization is a line of future work that has natural analogues in optimization of communication systems and fault-tolerant computation (Richardson & Urbanke, 2008).

### Acknowledgment

We appreciate discussion with and valuable suggestions from Razan Baltaji, Akhil Bhimaraju, and Moulik Choraria.

This work was supported in part by National Science Foundation grant PHY-2112890 and by DARPA grant "Modeling and Measuring Scientific Creativity".

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

# A  SOLVING EQUATION 1: MAXIMIZING CONCEPT LEARNING UNDER COMPUTE BUDGET CONSTRAINT

## A.1  A BRIEF SUMMARY OF BELIEF PROPAGATION DECODING OF LDPC CODES UNDER ERASURE

Low-density parity check (LDPC) codes are a family of error-correction codes, whose noisy codewords can be decoded in a computationally efficient manner using belief propagation. Before getting into deriving the probability that a concept is learnt from text pieces, we provide a very short summary of belief propagation decoding of LDPC codes when codeword symbols are corrupted by erasure. An LDPC code can be graphically represented by a Tanner graph, which is a bipartite graph with a set of variable nodes (codeword symbols) and check nodes (parity checks). Each codeword satisfies all the parity checks. Given a degree distribution pair (for variable and check nodes), there is a channel noise

threshold $\epsilon^*$ above which the decoder fails to decode the transmitted codeword. Consider a noisy version of a transmitted codeword with $\epsilon < \epsilon^*$ fraction of the symbols are erased. Belief propagation decoding starts by finding a check node where all except one symbol are recieved correctly (not erased). Then the erased symbol is determined as the one satisfying the parity. The next iteration starts by finding another check node with only one erased codeword symbol. This process continues until either all the codeword symbols are decoded or the decoder gets stuck with no parity checks containing only one erased symbol. The latter is declared as a decoding failure.

## A.2 COMPUTING $\Pr\{r \notin \mathcal{R}_+ | R, T\}$

Learning concepts from texts by the peeling process described in Section 2.3 is identical to belief propagation decoding of an LDPC code when the channel noise is erasure. To see this, treat $R$ concepts as erased codeword symbols (subset of variable nodes), and $T$ text pieces as parity checks. To obtain one-to-one correspondence, we need un-erased symbols (the remaining subset of variable nodes). Therefore, we choose (arbitrarily) a channel noise parameter $\epsilon \in (0, 1)$, add $\frac{1-\epsilon}{\epsilon} R$ nodes (dummy nodes) to the set of variable nodes, and treat them as un-erased symbols. Next, add edges between every pair of dummy variable node and a parity check node with probability $p = \frac{d_t}{R}$. Consequently, the degree distribution of the parity check nodes (text pieces) is modified, i.e., its degree distribution is binomial with parameters $R/\epsilon$ (instead of $R$) and $d_t/R$, but the degree distribution of variable nodes remains unchanged. Let us call the resulting parent graph $\widetilde{G}_1$[4] (see Figure 5) with the following text and concept degree distributions,

$$\widetilde{P}_R = \text{Binomial}(R/\epsilon, p), \text{ and} \tag{14}$$

$$\widetilde{L}_T = L_T = \text{Binomial}(T, p), \tag{15}$$

respectively. Here, for a compute budget $C$, we set $T = \frac{C}{6\varsigma\tau R}$.

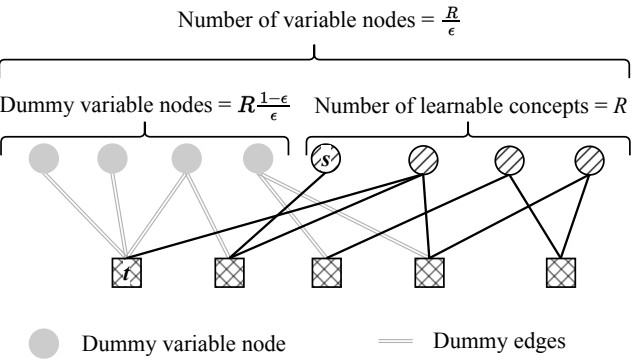

Figure 5: Bipartite graph $\widetilde{G}_1$.

In belief propagation decoding (peeling) of a codeword affected by erasures, the post-decoding bit erasure rate depends only on the residual graph consisting only variable nodes corresponding to erased symbols, parity checks connecting those variable nodes, and edges between them. Therefore, the post-decoding bit erasure rate is invariant to the choice of $\epsilon$.[5] Therefore, we can make the following equivalence between concept learning and bit erasure rate:

$$\Pr\{r \notin \mathcal{R}_+ | R, T\} = \frac{P_{b, \lambda_T, \widetilde{\rho}_R}}{\epsilon}, \tag{16}$$

where $P_{b, \lambda_T, \widetilde{\rho}_R}$ is the post-decoding bit erasure rate, and $\lambda_T(x) = \frac{L'_T(x)}{L'_T(1)}$ and $\widetilde{\rho}_R(x) = \frac{\widetilde{P}'_R(x)}{\widetilde{P}'_R(1)}$ are variable and check node degree distributions from edge perspective, respectively. To compute

---

[4]In this section, we omit superscript $(C)$ in $\widetilde{G}_1^{(C)}$ for brevity.

[5]Here we choose $\epsilon = 0.5$ (instead of close to 0 or 1) for numerical convenience.

$P_{b,\lambda_T,\widetilde{\rho}_R}$ we need the following ingredients: degree distributions $\lambda_T$ and $\widetilde{\rho}_R$, decoding threshold $\epsilon^*$, and scaling factors $\nu^*$ and $\alpha$ which depend on degree distributions. Degree distribution of text pieces from the node perspective is

$$P_R(x) = \sum_i \binom{R}{i} p^i (1-p)^{R-i} x^i, \tag{17}$$

$$\widetilde{P}_R(x) = \sum_i \binom{R/\epsilon}{i} p^i (1-p)^{(R/\epsilon)-i} x^i, \tag{18}$$

which gives the following text degree distribution from the edge perspective:

$$\widetilde{\rho}_R(x) = \frac{\widetilde{P}'_R(x)}{\widetilde{P}'_R(1)} = \frac{\sum_i i \binom{R/\epsilon}{i} p^i (1-p)^{(R/\epsilon)-i} x^{i-1}}{\sum_i i \binom{R/\epsilon}{i} p^i (1-p)^{(R/\epsilon)-i}}. \tag{19}$$

Noting that $i\binom{R/\epsilon}{i} = R\binom{R/\epsilon-1}{i-1}$ we obtain the degree distribution of text pieces from edge perspective:

$$\widetilde{\rho}_R(x) = \frac{\sum_{j=0}^{(R/\epsilon)-1} \frac{R}{\epsilon} p \binom{R/\epsilon-1}{j} p^{i-1} (1-p)^{(R/\epsilon)-i} x^{i-1}}{\frac{R}{\epsilon} p} \tag{20}$$

$$= (px + (1-p))^{\frac{R}{\epsilon}-1}. \tag{21}$$

Similarly, the degree distribution of concepts (remains unchanged for a fixed $R, T$) from the edge perspective is

$$\lambda_T(x) = (px + (1-p))^{T-1}. \tag{22}$$

Next the belief propagation decoding threshold $\epsilon^*$ is obtained from its fixed point characterization (Richardson & Urbanke, 2008, Section 3.12):

$$\epsilon^* = \inf\{\epsilon \in [0,1] : x = f(x, \epsilon) \text{ has a solution in } x \in (0,1]\}, \tag{23}$$

where $f(x, \epsilon) = \epsilon \lambda_T(1 - \widetilde{\rho}_R(1-x))$, and the critical point $x^*$ satisfies $x^* = f(x^*, \epsilon^*)$.

From finite-length scaling law of error rates in belief propagation decoding (Richardson & Urbanke, 2008, Section 3.23), we have the following (approximate) closed-form expression for post-decoding bit erasure rate:

$$P_{b,\lambda_T,\widetilde{\rho}_R} \approx \nu^* Q\left(\sqrt{\frac{R}{\epsilon}} \frac{(\epsilon^* - \epsilon)}{\alpha}\right), \tag{24}$$

where $\nu^* = \epsilon^* \, L_T(1 - \widetilde{\rho}_{\mathcal{R}}(1-x^*))$, $Q(\cdot)$ is the complementary standard Gaussian cumulative distribution function, and the scaling parameter $\alpha$ is given by (Richardson & Urbanke, 2008, Section 3.23)

$$\alpha = \left(\frac{\rho(\bar{x}^*)^2 - \rho((\bar{x}^*)^2) + \rho'(\bar{x}^*)(1 - 2x^*\rho(\bar{x}^*)) - (\bar{x}^*)^2\rho'((\bar{x}^*)^2)}{L'_T(1)\lambda_T(y^*)^2 \rho'(\bar{x}^*)^2} + \tag{25}$$

$$\frac{(\epsilon^*)^2 \lambda(y^*)^2 - (\epsilon^*)^2 \lambda_T((y^*)^2) - (y^*)^2(\epsilon^*)^2 \lambda'_T((y^*)^2)}{L'_T(1)\lambda(y^*)^2}\right)^{1/2}, \tag{26}$$

where $x^*$ is the unique critical point, $\bar{x}^* = 1 - x^*$, and $y^* = 1 - \widetilde{\rho}_R(1-x^*)$.

## B   CALCULATION OF $P_{e,train}$

We assume that the training error has contributions from two terms. The first contribution is from fraction of text pieces connected to unlearnt concepts, and the second term has an inverse relationship with the model size (equivalently, the number of learnable concepts). Since we do not know the precise functional form of the latter, we focus on the former, which gives a lower bound on the training error.

The probability that a text piece is connected to an unlearnt concept is equivalent to finding the probability that a text piece is connected to the set of unlearnt concepts at least twice (also called the stopping set in iterative decoding of LDPC codes), i.e.,

$$p_s := \Pr\left(|\{e_{tr} \in G_1^{(C)}\}_{r \in \mathcal{R}_-}| \geq 2\right), \text{ for any } t \in \mathcal{T}, \tag{27}$$

The training error is bounded below as follows.

$$P_{e,train} \geq p_s = \Pr\left(|\{e_{tr} \in G_1^{(C)}\}_{r \in \mathcal{R}_-}| \geq 2\right), \text{ for any } t \in \mathcal{T}, \tag{28}$$

$$= \sum_{k=2}^{R} \Pr\left(\text{degree}(t) = k, \{|\{e_{tr} \in G_1^{(C)}\}_{r \in \mathcal{R}_-}| \leq 1\}^c\right), \tag{29}$$

$$= \sum_{k=2}^{R} \binom{R}{k} p^k (1-p)^{R-k} \left(1 - (1-P_b)^k - kP_b(1-P_b)^{k-1}\right), \tag{30}$$

where the edge probability $p = d_t/R$ and $P_b = \epsilon^{-1} P_{b,\lambda_T,\widetilde{\rho}_R}$. The last equation simplifies to:

$$P_{e,train} \geq 1 - \left(1 - \frac{d_t P_b}{R}\right)^R - d_t P_b \left(1 - \frac{d_t P_b}{R}\right)^{R-1}, \tag{31}$$

which is obtained by computing the expectation of each of the three terms within the summation in equation 30 and substituting $p = d_t/R$. Further using the approximations $(1-x)^n \approx 1 - nx$ and $R - 1 \approx R$ for large $R$, the training error is bounded below as $P_{e,train} \gtrsim d_t^2 P_b^2$.

## C  CALCULATION OF $p_l$

Recall that $p_l$ is the probability that the composition of a pair of skills in level $l$ is seen at least $\eta_l$ times in the training data. For a fixed pair of skills $(s_1, s_2)$, the probability there is a path between the pair of skills through some pair of concepts $(r_1, r_2)$ is

$$\Pr(s_1 - r_1 - r_2 - s_2) = \Pr(s_1 - r_1, r_1 - r_2, r_2 - s_2),$$
$$= \Pr(s_1 - r_1)\Pr(r_1 - r_2)\Pr(r_2 - s_2),$$
$$= \frac{1}{S^{(l)}}\left(1 - \left(1 - \frac{d_t^2}{R^2}\right)^T\right)\frac{1}{S^{(l)}} =: p_{rr},$$

where the second inequality is due to independence of $s_1 - r_1$, $r_1 - r_2$ and $r_2 - s_2$. Let $X$ be a random variable indicating the number of distinct paths $s_1 - r_1 - r_2 - s_2$ between $s_1$ and $s_2$. Now, $\Pr(\text{composition of}(s_1, s_2) \text{ in training data}) =: p_l$ is

$$p_l = \Pr(X \geq \eta_l, \text{all prerequisite skills of } s_1 \text{ and } s_2 \text{ are acquired}),$$
$$\geq \Pr(X \geq \eta_l)\Pr(\text{all prerequisite skills of } s_1 \text{ and } s_2 \text{ are acquired}).$$

Note that the total number of distinct paths between $s_1$ and $s_2$ equals the total number of concept pairs $(r_1, r_2)$ which is $\binom{R}{2}$, each with probability $p_{rr}$. Therefore, $X$ follows a binomial distribution, i.e., Binomial $\left(\binom{R}{2}, p_{rr}\right)$. From Chernoff's bound for binomial distribution, we obtain the following lower bounds:

$$\Pr(X \geq \eta_l) \geq \begin{cases} \left(1 - \exp\left(-\binom{R}{2}D_{KL}\left(\frac{\eta_l}{\binom{R}{2}}||p_{rr}\right)\right)\right) & \text{if } \eta_l \leq \binom{R}{2}p_{rr} \\ \frac{1}{\sqrt{8\eta_l\left(1 - \frac{\eta_l}{\binom{R}{2}}\right)}}\exp\left(-\binom{R}{2}D_{KL}\left(\frac{\eta_l}{\binom{R}{2}}||p_{rr}\right)\right) & \text{otherwise.} \end{cases} \tag{32}$$

In deriving the above lower bounds, the following versions of Chernoff's bounds are used:

$$F(k; n, p) \leq \exp\left(-nD\left(\frac{k}{n}||p\right)\right), \text{ and} \tag{33}$$

$$F(k; n, p) \geq \frac{1}{\sqrt{8n\frac{k}{n}\left(1 - \frac{k}{n}\right)}}\exp\left(-nD\left(\frac{k}{n}||p\right)\right), \tag{34}$$

where $F(k; n, p) = Pr(Z \leq k)$ with $Z \sim \text{Binomial}(n, p)$.

The probability of acquiring prerequisite skills of both skills $s_1$ and $s_2$ is (assuming $R \gg \sigma_l$),

$$\Pr(\text{all prerequisite skills of } s_1 \text{ and } s_2 \text{ are acquired}) \geq \Pr(\text{all } \sigma_l \text{ prerequisites} \in \text{GCC})^2,$$
$$= \gamma_{l-1}^{2\sigma_l}.$$

