# OpenReview forum: "An Information Theory of Compute-Optimal Size Scaling, Emergence, and Plateaus in Language Models"
_ICLR.cc/2025/Conference — Submitted to ICLR 2025_

### Official Review · Reviewer_kET7 · 2024-10-21

**Soundness:** 3
**Presentation:** 2
**Contribution:** 2
**Rating:** 6
**Confidence:** 4

**Summary:**

This paper provides an unified mathematical framework to explain three phenomena with increasing size of language models: compute-optimal size scaling, emergent capabilities, and performance plateauing. This framework is proposed based on the notation of learning as two levels: (1) a set of concepts are learnt from a set of texts;  (2) learning concepts enables the language model to acquire skills. The authors prove the compute-optimal (Chinchilla) scaling rule based on non-asymptotic information-theoretic tools to the bipartite graph between texts and concepts, and explain the emergence and plateauing phenomena based on the density of connections in the skill-graphs.

**Strengths:**

This paper has some strengths:

+ The authors can explain three phenomena mentioned above by using their unified framework.
+ The trick to solve the optimisation problem (1) by considering the peeling process of learning concepts from texts to be identical to belief propagation decoding of an LDPC code when the channel noise is erasure is interesting.

**Weaknesses:**

This paper contains some weaknesses:
+ Some mathematical approximations should be carefully re-thought (please see my question below).
+ The gap between empirical excess entropy and its lower bound are too big.

**Questions:**

The LHS of (4) is an expression that does not depend on $\epsilon$. However, the approximation expression in the RHS of (4) depends on $\epsilon$. How can we explain about this approximation? Can $\epsilon$ take arbitrary value as mentioned in the proof  in Appendix A.2?

---

> ### Author Response · Authors · 2024-11-22
>
> We thank the reviewer for insightful comments. The responses to reviewer's questions/concerns are provided below.
>
> > The gap between empirical excess entropy and its lower bound are too big.
>
> We thank the reviewer for the observation. Indeed, this could suggest that either there is a significant scope for exploring better neural architectures, or the bound is loose, the investigation of which is left for future work. Please note that the excess entropy scaling is not the main message of the paper. A main contribution of our paper is obtaining the Chinchilla size scaling rule described in the previous subsection, i.e., Section 3.1.
>
> > The LHS of (4) is an expression that does not depend on $\epsilon$. However, the approximation expression in the RHS of (4) depends on $\epsilon$. How can we explain about this approximation? Can take arbitrary value as mentioned in the proof in Appendix A.2?
>
> We thank the reviewer for the question about technical detail. As the reviewer rightly pointed out, the right side of (4) should be invariant to $\epsilon$. This is true, but it is not very apparent. The argument is the following (which is provided in Appendix A.2): Consider an instance of bipartite graph $\tilde{G}_1$. At every peeling/decoding iteration, the final set of unlearned concepts (undetermined codeword bits after decoding) depends only on the residual graph (containing only the concepts yet to be learnt and adjacent text pieces) of that particular iteration. In the first peeling/decoding iteration, the residual graph is $G_1$. The choice of $\epsilon$ only affects the number of dummy variable nodes and associated edges; equivalently it only affects the constructed parent graph $\tilde{G}_1$, but not $G_1$ which is the original graph. Therefore, the probability that a concept is not learned after the peeling process (or equivalently, probability that a codeword bit is still erased after decoding conditioned on codeword bits erased before decoding) does not depend on the choice of $\epsilon$.
>
> Note that the quantities $\epsilon^*$, $\alpha$, $\nu^*$ vary with the value of $\epsilon$ such that $\epsilon^{-1} P_{b, \lambda_T, \tilde{\rho}_R}$ is unchanged for a fixed $(R, T)$.
>
> On a related note, a rigorous treatment of the evolution of degree distributions of residual graphs and iterative decoding as a Markov process is discussed in Amraoui et al.\ (2004) (https://arxiv.org/abs/cs/0406050).

---

> > ### Comment · Reviewer_kET7 · 2024-11-26
> >
> > Thank you very much for your responses to my questions. Your answers partially resolve my concerns. Hence, I raise my score to 6.

---

### Official Review · Reviewer_TZYr · 2024-10-22

**Soundness:** 2
**Presentation:** 2
**Contribution:** 2
**Rating:** 5
**Confidence:** 3

**Summary:**

The paper proposes an abstraction of the learning process of a large language model using graphs and analyze the behavior of certain dynamics on these graphs. The purpose is to draw a connection to certain observed phenomena of large language models such as the emergence of learned capabilities and the plateauing of performance with size-scaling.

I found the topic and the framework interesting and worthy of a discussion. However, there are many weaknesses in writing, presentation, and technical issues that makes it difficult to evaluate the paper’s contributions.

**Strengths:**

Motivation: The framework seems reasonable and diverse enough to model learning and acquisition of new skills. As the authors mentioned, it may be possible to incorporate into the framework additional considerations such as the quality of training data. It would be interesting to discover what can we learn in the training of language models due to this analysis.

The topic and theoretical framework are related to other recent works, hence it seems like this topic has created enough interest in the community.

Using tools developed in coding theory and random graph theory in probability seems to have great potential in this context. The authors say “Our work takes a step in grounding empirical phenomena observed in size scaling of language models on a rigorous mathematical footing.” I would put it differently: ``The work proposes a graph-based model for learning skills from texts that appears relevant to the process of training a large language model. The dynamics of message passing under an asymptotic scaling of the model’s size experience phenomenon akin to those empirically observed in this training process of a large language model, such as the emergence of abilities and the plateauing of performance.’’

**Weaknesses:**

Background and Motivation.
It is unclear what parts of the framework have been considered in previous works. For example, did other work also consider hierarchical skills? Did previous works also try to analyze message-passing dynamics but perhaps using different tools? How significant is the automatic selection of scaling law from the framework compared to previous works? What is the rationale that N (#of parameters) is proportional to R (# of skills)? What is the rationale for studying a limited computed budget and thus the tradeoff N T < C? Do we see such a tradeoff in training language models?

Also, in the abstract and introduction, please be clear whether ‘size’ means the number of parameters or the size of training data.

The connection to language models is not immediate and it appears that one can replace the language model with any form of a learning machine, like a human. Consequently, I’d expect in this line of work at least some discussions on studies on learning and education in general. Also related: Line 102 says that a language model “chooses to learn.” This seems like a poor choice of words because it is unclear what is the choosing mechanism. Another similar poor choice of word or issue with the motivation is L245: “The goal of the language model is to learn as many concepts as possible…” Of course, actual training in a language model does not directly involve the concept of “concepts”.

The authors say that they use tools from information theory but this is inaccurate because they use statistical mechanics ideas from graph-based coding (Richardson & Urbanke 2008) and random graph theory. The terms information theory and especially non-asymptotic information theory are misleading.


Analysis.
In general, I found it difficult to follow almost all technical derivations due to missing details. Examples:
What are the {Pi} s in L124?
What is the “matching condition” in L316
What is r in Eq 2 ? I don’t see how this equation is equivalent to the problem in Eq 1.
What are r and epsilon in Eq 3?
What is epsilon in Eqs 7 & 8?
What is the origin of the name “post-decoding bit erasure rate”?
What is “excess entropy”? Why do we care about bounding it from below?

The paper mentions compute-optimal scaling but I did not find a clear definition.
The paper does not explain what the author calls the ``decoding process’’.

Proposition 1 appears to be flawed. It is not true that if neither R/T = o(1) nor T/R = o(1) hold then R/T → constant (the text actually says “R/T must be a constant”, which is grossly incorrect). As a counterexample, take R = sin(C), T=cos(C). To my understanding, the author quotes this proposition as one of the main contributions of their work, so this needs correction. This proposition is also unclear because the notion of “compute-optimal performance of a language model” has not been well-defined earlier.

**Questions:**

Some suggestions:
Please specify in the setup that the skills graph obeys the  Erdos-Renyi random graph model, as you later claim in L346.

Paragraphs in Lines 210-215 are related to the high-level discussion of “what is a skill” and therefore should be part of the exposition.

The study would have a much better case if the authors could show measurable quantities like accuracy and the number of learned skills obtained from the model side by side with those obtained from actual language models.

---

> ### Author Response · Authors · 2024-11-22
> **Response 1: part 1**
>
> We thank the reviewer for providing insightful feedback. We provide responses to reviewer's questions/concerns below and we have updated our manuscript accordingly.
>
> Note: Due to character limit, we split our response into two cells.
>
> > did other work ... hierarchical skills?
>
> One notion of hierarchical skills is considered in Liao et al (2024) in the context of fine-tuning a model. They consider two levels of skills - basic skills and domain-specific skills. To learn domain specific skills a language model needs domain-specific texts and basic skills. On the other hand, we do not consider fine-tuning. Instead, we modify/augment this graphical model with a hierarchy of skills inspired by cognitive architectures in Laird et al., 1987; Anderson 1993; Kieras and Meyer, 1997 (we discuss this between L059 to L066 in the introduction section of our paper), and skills are acquired from concepts and not directly from texts. To the best of our knowledge we are not aware of works discussing a graph-based model of hierarchy of skills and acquiring skills from concepts in the same form as in our paper.
>
> > Did previous works ... message-passing ... different tools?
>
> Yes. Thinking of language model training as a peeling process (message-passing) is borrowed from Liao et al. 2024. In Liao et al. 2024, emergence is explained through asymptotic analysis (however, their notion of emergence is different from ours as summarized in Section 2.5 of our paper). We start from the same message-passing dynamics, but our contribution is finitary analysis; we first prove the optimality of the Chinchilla scaling rule, and then explain emergence.
>
> > How significant ... automatic selection of scaling law ...?
>
> In previous works, the scaling law must be assumed a priori to explain emergence of skills. However, from our finite blocklength analysis, we prove the optimality of the Chinchilla compute-optimal size scaling. The goal is to first provide a mathematical explanation to the scaling law and then explain emergence of skills.
>
> > What ... rationale ... $N$ ... is proportional to $R$ (No. of skills)?
>
> We borrow the notion of skill-quanta from Michaud et al. (2023), and so the number of concepts (R) a language model can learn is proportional to the model size (number of model parameters). The model size determines its capacity in terms of the number of concepts learnt. The paragraph ``parameter scaling" in Section 2 of Michaud et al. (2023) describes the relationship between skill quanta and number of model parameters.
>
> > What ... rationale ... $N T < C$ ... language models?
>
> The tradeoff between number of parameters of language models and dataset size due to compute budget is empirically studied in papers such as Hoffmann et al. 2022, Kaplan et al., 2020, etc. The rationale is: for a fixed compute budget, one can either train a large model with small dataset size or a small model with a large dataset size. For a language model with a large number of parameters, large number of FLOPs are required for each gradient update. Studying the trends for smaller compute budgets provides a thumb rule to determine (model size, dataset size) pair for larger compute budgets.
>
> > Also, ... whether ‘size’ means ... data.
>
> We have now indicated that model size means the number of model parameters in the introduction - ``so the number ... model size (number of model parameters)."
>
> > The connection to language models ... learning and education in general.
>
> Agree. The framework is not restricted to transformer-based language models and may characterize other learning paradigms. In the introduction we highlight this generality: ``Moreover, although ... tied to Transformer-based language modeling architectures (Yu et al., 2023a), it can describe ... different learning paradigms (Yu et al., 2023b)." Note that the paper "Information Lattice Learning" by (Yu et al., 2023b), is itself a human-like learning approach, where lattice construction using universal priors resembling human innate cognition --- the Core Knowledge priors (Spelke and Kinzler, 2007), and learning within the lattice to find optimal rules that best explain the signal.
>
> > Also related: ... poor choice of words ...
>
> We have modified the sentences in section 2.1 - ''A set of tokens ... a language model can learn a set of concepts ... Note that we consider ... Hoffmann et al. (2022)." We have modified the sentence in L245 to ``The goal ... under the compute budget constraint $C$".
>
> > The authors say ... non-asymptotic information theory are misleading.
>
> Please note that we think of modern coding theory as a branch of information theory, rather than narrowly defining information theory as only Shannon theory. Indeed, Shannon theory is only one topic among many (including modern coding theory) that appear in venues such as the IEEE Transactions on Information Theory. In particular, all the main papers on LDPC codes and their analysis appear in the IEEE Transactions on Information Theory.
>
> (continued)

---

> > ### Author Response · Authors · 2024-11-22
> > **Response 1: part 2**
> >
> > > ANALYSIS.
> > > What are the Pi s in L124?
> >
> > $P_i$ in L124 is $\Pr(X = i)$, where $X$ is a Binomial random variable.
> >
> > > What is the “matching condition” in L316 What is r in Eq 2? I don’t see how this equation is equivalent to the problem in Eq 1.
> >
> > We apologize for the lack of clarity. This is a coding theory terminology---the matching condition is used to construct capacity-achieving degree distributions. However, we are only interested in an upper bound of the decoding threshold $\epsilon^*$. So, we need not invoke this term. Therefore, we have reworded the sentence along with specifying the Section number in the reference for readers interested in understanding more about the upper bound of decoding threshold: ``The upper bound on the decoding threshold is given by (see (Richardson and Urbanke, Section~3.14.4))."
> >
> > > What are r and epsilon in Eq 3? What is epsilon in Eqs 7 and 8?
> >
> > r is a concept in $\mathcal{R}$, which is introduced in L121: "... indicates that the language model can learn concept r from text t." epsilon in Eq 3 is an arbitrarily chosen value in (0, 1). We have added a footnote "$\epsilon \in (0, 1)$ can be chosen arbitrarily ... See Appendix A.2 for details."
> >
> > > What is ... “post-decoding bit erasure rate”?
> >
> > A brief summary of belief propagation decoding of LDPC codes and the definition of post-decoding bit erasure rate, and how it relates to probability that a concept is learnt is provided in Appendix A.
> >
> > > What is “excess entropy”? Why do we care about bounding it from below?
> >
> > The cross-entropy loss is the sum of the entropy of the ground truth distribution and excess entropy (please refer to Equations (5) and (6) in Arora and Goyal, 2023 (https://arxiv.org/pdf/2307.15936) for details on excess entropy). Excess entropy is the KL divergence term in the loss expression. A lower bound on excess entropy reveals the gap between the empirically observed training loss and quantifies the room for improvement. If the gap is small, then it suggests that search for better model architectures would only yield marginal improvements, whereas if the gap is large, then it either suggests that there is still room for improving model architectures, or that there is a need for exploring better (larger) theoretical lower bounds.
> >
> > > The paper ... compute-optimal scaling ... does not explain ... ``decoding process’’.
> >
> > We apologize for the lack of clarity. The idea of compute optimal size scaling as described in Hoffmann et al 2022 (arxiv.org/abs/2203.15556) is the following: For every compute budget C measured in FLOPs, there is a corresponding (model size, dataset size) pair (N, D) with minimum training loss (compute-optimal pair). It was empirically found that both compute-optimal N and D scale as $C^{1/2}$ (approximately). We have revised the statement of Proposition 1, which describes the idea of Chinchilla optimal size scaling formally.
> >
> > The equivalence between learning concepts from texts and the iterative decoding process (belief propagation or message passing) of LDPC codes is explained in Appendix A.1.
> >
> > > Proposition 1 appears to be flawed. ... has not been well-defined earlier.
> >
> > We thank the reviewer for careful review of the proof and pointing out the flaw in the argument. We have now revised the proposition statement by asserting that N and D (equivalently, R and T) obey polynomial scaling, and prove that the exponents must be equal, i.e., $\alpha = \beta = 1/2$.
> >
> > > Please specify ... Erdos-Renyi ... “what is a skill” .. should be part of the exposition.
> >
> > We thank the reviewer for the suggestion. We have added the following sentence in Section 2.5 - ``As we will see in ... the skill graph with nodes $\mathcal{S}^{(l)}$ is an Erdos-Renyi (ER) random graph with ... connected component.''
> >
> > > The study would have a much better case if the authors could show measurable quantities like accuracy and the number of learned skills obtained from the model side by side with those obtained from actual language models.
> >
> > We thank the reviewer for the suggestion. This can indeed be a separate empirical work to evaluate the theory described in this paper. One can use evaluation frameworks like ``Skill-Mix: A Flexible and Expandable Family of Evaluations for AI models'' by Yu et al., 2023a, which presents enumerable skills and elaborate evaluation methodology (in Appendix A of their paper), and which has been previously been used to demonstrate the validity of the broad mathematical approach adopted here.
> >
> > On a related note, we would like to emphasize that the the Chinchilla size scaling rule obtained from our framework matches closely with the empirically observed values (for Chinchilla model in Hoffmann et al., 2022) as highlighted by "*" markers in Figure 3(a) in our paper.

---

> ### Comment · Reviewer_TZYr · 2024-11-24
>
> Thank you for your response.
>
> I have no further questions or comments.
>
> I have increased my initial rating from 3 to 5 because the authors have partially addressed my concerns.

---

### Official Review · Reviewer_5UXU · 2024-10-27

**Soundness:** 1
**Presentation:** 2
**Contribution:** 1
**Rating:** 3
**Confidence:** 5

**Summary:**

The authors proposed a theory for scaling in neural networks via a connection to LDPC code. The authors then claim that the theory leads to explanation of scaling, emergence, and plateaus in language models.

**Strengths:**

The goal of the paper is ambitious. A unifying theory that can help explain the scaling behavior will be quite nice.

**Weaknesses:**

Though the effort is appreciated, and there might be some merit to the overall approach, I feel the overall theory is not set up in sufficiently rigorous manner.

1. Reading the paper gives me the feeling that authors are promising too much, but nothing has been theoretically established carefully. First of all, the framework is given in a very vague manner, with the basic definitions for text, concepts, and skills missing. Without rigorous definitions, we cannot come up with simple examples to verify the theory. For example, if I'm given a text sequence, how do I determine if it is a text, a concept, or a skill? Are they all limited by the representation and the length? The theory is shaky when we cannot even answer these basic questions.

2. There seems to be less connection to ML than to LDPC. If we simply replace the text, concept, and skill nodes in the graph with variable and factor nodes in the original LDPC, then the claim holds trivially. This is an extremely weak connection, in names only, without any deeper insights.

3. There are no relevant experimental results for language models at all in the paper. For a theory to hold, we must have some kind of verification with the ML models. The paper reads like this: there are some observations in ML, and it happens that similar effects also occur in other settings, so that theory will fit. This might be a good starting point to contemplate and develop deeper research, but scientific/engineering research cannot stop here.

4. The big concern for me is that the theory is so vaguely defined that it is impossible to come up with experiments to verify whether it is true or not. This is a rather dangerous territory, as it does not follow the standard scientific research approach, and is more like a belief/religion in its current form.

**Questions:**

Simple questions as mentioned earlier:
1. How to define text, concepts, and skills, and given some test or bit sequence, how do we determine which category it falls into?
2. What experiments can be used to verify the theory?

---

> ### Author Response · Authors · 2024-11-22
>
> We thank the reviewer for feedback and we aim to provide clarity here.  Please note that experimental validation for our basic theoretical model is, as detailed below, directly provided by prior work that we cite.\\
>
> We follow the definition of a text piece from Arora and Goyal, 2023 (https://arxiv.org/abs/2307.15936). The text piece is a set of tokens, for example, a fixed number of sentences or a fixed number of tokens.
>
> We consider a concept to be an abstract quantity that a model learns. In transformer-based language models, concepts could be interpreted as clusters in self-attention dynamics according to Geshkovski et al.\ (https://neurips.cc/virtual/2023/poster/71198). On the other hand, in a more human-interpretable information lattice learning (Yu et al., 2023b), concepts could be interpreted as abstract hierarchical rules extracted from the data.
>
> Skills can be enumerable human-interpretable notions according to Yu et al., 2023a (https://iclr.cc/virtual/2024/poster/18931). Some of the skills listed in the paper are: self serving bias, accident, complex question, red herring, metaphor, spatial reasoning, and modus ponens.
>
> The paper "SKILL-MIX: A flexible and expandable family of evaluations for AI models" by Yu et al., 2023a (https://iclr.cc/virtual/2024/poster/18931) provides strong experimental results on the validity of text-pieces and skills. In that paper, a text piece is made up of 3 sentences, and they consider a set of 101 skills (some of which are listed above). However, the authors suggest that the number and definition of skills can be updated: "... adding new skills to our list, or by switching to more difficult or specialized skills, or to topics that are rarer or more specialized."
>
> Designing experiments to test our framework is indeed feasible. For instance, taking inspiration from cognitive studies (Bloom's Taxonomy or cognitive hierarchy theory) one can enumerate and construct a hierarchy of skills. Then define tasks (for example, text generation, cloze questions, etc) involving a specific subset of skills from multiple skill levels. Next, use the SKILL-MIX framework to evaluate a language model's performance (for example quality of generated text, accuracy, etc) on the task. One can evaluate the performance for increasing compute budgets. To evaluate the learning of concepts, one could investigate how clustering behavior in self-attention dynamics evolves with scaling.
>
> Since the mathematical underpinnings of empirical phenomena are not studied extensively, what we are doing is explaining several empirical observations using theoretical properties of experimentally-verified mathematical models.

---

> > ### Comment · Reviewer_5UXU · 2024-11-29
> >
> > I would like to thank the authors for their reply, which provided more context for their motivation.
> >
> > However, I remain unconvinced by the formulation and the claimed result. The preprint by Arora and Goyal, 2023 was never officially reviewed, and I have a similar reservation about the "text piece" given there, particularly in connection with other concepts in the current work. For example, how long should a "text piece" be, and if it is sufficiently long, is it already a "concept"? Does the length impact the claimed results? The work of Yu et al., 2023a was to propose an evaluation framework, but the relevant definitions were not made formal in a mathematical manner. The work by Geshkovski et al. is a token-level mathematical study that is not directly related to the high-level concepts needed in this submission. The task of making the concepts of "text, concepts, and skills" mathematically rigorous is therefore a key difficulty the authors must overcome first. For example, how do we define "skill" or whether "a skill is learned" in the proposed framework? Does there need to be a formula that a transformer must be able to compute to a certain accuracy for it to be called a skill? If not, what other criteria? Such a definition would need to eventually manifest in the theoretical results developed, as otherwise, we can always define some meaningless skills that either would never be possible or others that can be trivially learned. In the evaluation framework by Yu et al., these were not an issue since they can simply evaluate the "skills" using data or tests, but in a mathematical framework, we cannot use data or numerical tests, and therefore some mathematical definitions must be provided at the very beginning.
> >
> > The authors claim that "Designing experiments to test our framework is indeed feasible", and I believe this is an important component that must be included so that the readers can verify the claims. The authors are strongly encouraged to carefully design a set of experiments to test the theory.
> >
> > In summary, my main concerns remain, and I'll have to keep the score.

---

### Official Review · Reviewer_x1Em · 2024-10-30

**Soundness:** 4
**Presentation:** 3
**Contribution:** 3
**Rating:** 8
**Confidence:** 4

**Summary:**

This paper present a very interesting theory on large language model scaling laws using ideas from information theory, random graphs, and low-density parity-check codes. By drawing similarities between concept-text graphs, skill-concept graphs to Tanner graphs in low-density parity-check codes, the paper provides a theoretical explanations of the phenomenon observed in large language models, including compute-optimal scaling law, scaling law of excess entropy, emergence, and plateauing. The theoretical predicted scaling laws are compared with the empirical laws observed in real-world language models.

**Strengths:**

*The paper presents a creative way of using existing theories in low-density parity-check codes and random graphs to solve large language model analysis problems.
*It is convincing that one theory can be used to explain multiple phenomenon in large language model scaling.
*The paper is of good quality, the theoretical analysis are solid.
*The paper is well-written.
*The theory is of significance, because understanding the scaling laws of large language models may provide further guidance for future model training and scaling.

**Weaknesses:**

*I think one assumption the paper makes is that the peeling process has stopped after the model training. Nowadays, many language models are only trained with several epochs. Can we always assume that the peeling process have already stopped after the training?
*Not necessary in this paper, but it would be good to have more numerical experiments that this peeling process indeed happens in training.
*The paper assumes that the graphs can be randomly generated, so that we have particular binomial degree distributions. Is it possible to verify indeed this is the case in the real-world. Do the theory in the paper also hold for other degree distributions.

**Questions:**

*Some minor issues, in appendix C, should the value in equation 29 is only approximately equal to the value in equation 30? Also, the approximation $(1-x)^n = 1-nx$ should be tight with some additional conditions. In this case, this approximation is tight, only when $p_b$ is also small.
*In line 759, Chernoff's bounds have multiple forms. One reference should be given here.

---

> ### Author Response · Authors · 2024-11-22
>
> We are glad the reviewer finds our theory to be very interesting! We thank the reviewer for insightful comments. The responses to reviewer's questions/concerns are provided below and we have updated our manuscript according to the comments.
>
> > I think one assumption... Can we always assume that the peeling process have already stopped after the training?
>
> We thank the reviewer for the question. However, the physical meaning of continuing the peeling process after training is not very clear. We request the reviewer to clarify this question further.
>
> > Not necessary in this paper, but ... more numerical experiments ... peeling process indeed happens in training.
>
> We thank the reviewer for the suggestion. We would like to highlight that the model training as a peeling process is borrowed from Liao et al.\ (https://arxiv.org/abs/2404.07009); our contribution is finitary analysis to prove the optimality of the Chinchilla size scaling rule (and also explaining true emergence and plateauing). Moreover, please note that one need not think that peeling directly models the LLM training processs; instead this is an abstraction operating at a semantic level similar to the textpiece-skill approach of Arora and Goyal 2023 (https://arxiv.org/abs/2307.15936) that is also an abstraction at the semantic level. We agree that numerical experiments on how LLMs learn during training could be an interesting separate line of work.
>
> > The paper assumes ... Do the theory in the paper also hold for other degree distributions.
>
> The theory holds for other degree distributions too. However, computing the post decoding bit erasure rate $P_{b, \lambda_T, \tilde{\rho}_R}$  for general degree distributions is not straightforward (with reference to Appendix A.2). In particular, it is challenging to find the degree distribution $\tilde{\rho}_T$ of the parent graph $\tilde{G}_1$ such that the check nodes (text pieces) of the residual graph has the degree distribution $\rho_T$. Considering general degree distributions could be one of the potential future extensions.  In fact optimizing degree distributions would describe the best kinds of training datasets or model architectures.
>
> > Some minor issues, in appendix C, ... when $p_b$ is also small.
>
> Equation 30 is exactly equal to the Equation 29, which is obtained upon simplification. There were a couple of typos in Equation 29 and 30, which we have now corrected, apologies. As rightly pointed out by the reviewer, we can use the approximation only when $P_b$ is small. In our case, it turns out that $P_b$ is indeed very small, in particular $P_b \ll 1/d_t$. In the figures we have used $d_t = 6$ and . However, please note that in Figure~3(b) we plot the exact expression in Equation 30 (not the approximate expression in Equation 7 or 8).
>
> > In line 759, Chernoff's bounds have multiple forms. One reference should be given here.
>
> We thank the reviewer for pointing out. We have added ``In deriving the above lower bounds, the following versions of Chernoff's bounds are used: ... Binomial(n,p)."

---

### Meta-Review · Area_Chair_qsTL · 2024-12-23

**Metareview:**

**Summary:**
This paper proposes a heuristic, phenomenological model which is claimed to be able to reproduce several scaling phenomena observed in language models. The model is inspired by peeling decoders of low-density parity-check (LDPC) error correcting codes applied on an erasure channel, as well as percolation on random graphs.

**Strengths:**
The topic is interesting and worthy of discussion. The objective posed in this paper is ambitious. The proposed model is successful in demonstrating three phenomena regarding scaling observed in large language models.

**Weaknesses:**
- The presentation in this paper rather heavily relies on other books and papers, so that the reader may face difficulty in understanding the arguments.
- As Reviewer 5UXU mentioned, the proposed model is phenomenological and defined only vaguely. Although it may reproduce some of the characteristics of language models, we remain unassured whether the model be valid. A good theory would be able to predict what will happen *quantitatively* and/or to produce a hypothesis which will be verifiable/falsifiable via carefully-designed experiments. It seems, however, that the proposed model in this paper would lack either of the properties that a good theory should have.
- The authors claim that the proposed model is a simple unified framework, and yet explaining several different aspects of language models. In my view the proposed model is a mere composition of peeling decoder for LDPC and percolation on random graphs, where the peeling decoder part would account for the scaling rule, whereas the percolation part would reproduce emergence and plateauing as a consequence of percolation phase transition. These two parts are somehow independent of each other, in the sense that the percolation part would function rather independently of what would take place in the peeling decoder part beyond what proportion of concepts are to be learnt, so that I do not really think the proposed model a *unified* one.

**Reasons:**
Alghouth I think that this paper poses an ambitious goal and would be successful in putting a starting point for further elaboration, my evaluation on this paper is more aligned with the reviewers with negative evaluation, mainly because of the phenomenological and heuristic nature of the proposal, which would lack the ability to make a quantitative prediction or to pose a verifiable hypothesis for those three phenomena taken up in this paper.

Additional points:
- Line 127: $d_t/T$ → $d_r/T$
- Line 323: (the ratio of the) number of (skills → concepts) learnt
- Equation (6): The closing parenthesis in the numerator of the leftmost side should be made larger. $\frac{1+\delta}{1-\delta'}$ → $\frac{1+\delta}{1+\delta'}$.
- Line 454: note that $W_0(xe^x)=x$ for ($x<-1$ → $x\ge-1$)
- Line 458: $\sigma_l=0$ → $\sigma_1=0$
- Line 468: The phase transition mentioned here should be sharp but should not be discontinuous, so that saying it as "step" may be misleading.
- Figure 4 (a)(c)(d): It seems that the authors plot not the accuracy itself but the lower bounds of the accuracy given in equations (12) and (13).
- Independence of equation (4) of $\epsilon$: It would be true that the quantity of interest here is defined independently of $\epsilon$, and thus the end result (equation (4)) *should be* independent of $\epsilon$. On the other hand, the explicit formulas do not really appear so, so that it would be appreciated if a direct proof that the derived formula actually gives an $\epsilon$-independent quantity.

**Additional Comments On Reviewer Discussion:**

The rating/confidence scores  of the four reviewers were 8/4, 6/4, 5/3, and 3/5, exhibiting a large split. This split has not been resolved even after the author rebuttal and the discussion between the authors and the reviewers.

---

### Decision · Program_Chairs · 2025-01-22

Reject